# Periodontal Disease: A Risk Factor for Diabetes and Cardiovascular Disease

**DOI:** 10.3390/ijms20061414

**Published:** 2019-03-20

**Authors:** Daniela Liccardo, Alessandro Cannavo, Gianrico Spagnuolo, Nicola Ferrara, Antonio Cittadini, Carlo Rengo, Giuseppe Rengo

**Affiliations:** 1Department of Translational Medical Sciences, Federico II University of Naples, 80131 Naples, Italy; liccardo.daniela@gmail.com (D.L.); nicola.ferrara@unina.it (N.F.); antonio.cittadini@unina.it (A.C.); giuseppe.rengo@unina.it (G.R.); 2Center for Translational Medicine, Temple University, Philadelphia, PA 19140, USA; 3Department of Neurosciences, Reproductive and Odontostomatological Sciences, Federico II University of Naples, 80131 Naples, Italy; gianrico.spagnuolo@gmail.com; 4Institute of Dentistry, I.M. Sechenov First Moscow State Medical University, 119146 Moscow, Russia; 5Istituti Clinici Scientifici-ICS Maugeri S.p.A., 82037 Telese Terme (BN), Italy; 6Department of Prosthodontics and Dental Materials, School of Dental Medicine, University of Siena, 53100 Siena, Italy; carlorengo@alice.it

**Keywords:** periodontitis, inflammation, bacteria, cardiovascular disease, diabetes

## Abstract

Periodontitis is a chronic inflammatory disease, initiated by the presence of a bacterial biofilm, called dental plaque, which affects both the periodontal ligaments and bone surrounding teeth. In the last decades, several lines of evidence have supported the existence of a relationship between periodontitis and systemic health. For instance, as periodontitis acts within the same chronic inflammatory model seen in cardiovascular disease (CVD), or other disorders, such as diabetes, several studies have suggested the existence of a bi-directional link between periodontal health and these pathologies. For instance, people with diabetes are more susceptible to infections and are more likely to suffer from periodontitis than people without this syndrome. Analogously, it is now evident that cardiac disorders are worsened by periodontitis, both experimentally and in humans. For all these reasons, it is very plausible that preventing periodontitis has an impact on the onset or progression of CVD and diabetes. On these grounds, in this review, we have provided an updated account on the current knowledge concerning periodontal disease and the adverse effects exerted on the cardiovascular system health and diabetes, informing readers on the most recent preclinical studies and epidemiological evidence.

## 1. Introduction

Periodontitis is a multifactorial, chronic inflammatory disorder, that can lead, if untreated, to the non-reversible damage of supportive tissues (periodontal ligament, cementum and alveolar bone) surrounding the teeth with consequent tooth loss [1]. Importantly, one of the major determinants of the development and progression of periodontal disease is represented by an increased concentration of pathogenic bacteria, within the dental plaque, that activates a massive noxious immune response [2]. For instance, the augmented concentration of bacterial surface molecules, such as lipopolysaccharides (LPS), stimulates the production of inflammatory mediators and cytokines that, in turn, promotes the release of the matrix metalloproteinases (MMPs). These tissue-derived enzymes then participate in the extracellular matrix remodelling and bone destruction [3,4]. Importantly, recent studies have clearly proven that these deleterious effects are not only restricted to the oral cavity but can affect the overall health of an individual. For this very reason, research involving the systemic implication of periodontitis has grown exponentially [5]. Indeed, periodontal pathogens can destroy the epithelium of the periodontal pocket thus allowing the entry of noxious endotoxins and exotoxins into the bloodstream [2]. This process leads to bacterial dissemination and systemic infection, with a consequent rise in inflammatory response. For instance, periodontal pathogens have been detected in disparate tissues and organs of the cardiovascular system including human cardiac tissue, pericardial fluids, heart valves and in atherosclerotic lesions [6,7,8,9,10,11,12,13]. For these reasons, in the last decades, periodontitis has been associated with the onset of systemic disorders including cardiovascular disease (CVDs) and diabetes [14,15,16]. In this regard, two meta-analyses, by Janket et al. and Kofhader and colleagues respectively, have evaluated the potential correlation between oral disease and CVD, concluding that periodontal disease is a potential risk factor for cardiovascular (CV) events, such as stroke and coronary heart disease [17,18]. Furthermore, it has been previously demonstrated that individuals with periodontitis have a sensible increased risk of developing CVDs, including myocardial infarction, heart failure, peripheral artery disease (PAD), atherosclerosis and stroke [16,19].

Interestingly, the link between periodontitis and diabetes has also been pursued. Several studies suggest that this association is bi-directional [20]. Indeed, individuals with diabetes are more likely to develop periodontitis and those subjects with periodontitis, and diabetes, present a worse glycemic control [21,22]. Therefore, based on this premise, it is of paramount importance to inform health professionals about the consequence of diseases affecting the oral cavity in that these are potentially associated with a range of pathologic conditions. Thus, the aim of this review was to update clinicians and basic scientists about the current evidence (experimental and clinical) that supports the existence of the relationship between periodontal disease and CVDs.

## 2. Pathogenesis of Periodontitis

An imbalance of micro-organisms forming the dental plaque (dysbiosis) is a major triggering factor for chronic gingivitis and periodontitis [2,23]. In addition, periodontitis is associated with, and probably caused by, an altered dynamic interaction among specific subgingival microbes, host immune responses, hazardous environmental exposure and genetic factors [24]. To date, almost 800 different species of bacteria have been identified and characterized in human dental plaque. Of relevance, the putative pathogens include Gram-negative and -positive members, such as Treponema denticola, Tannerella forsythia, Prevotella intermedia, Agregatibacter actinomycetemcomitans, Campylobacter rectus, Eubacterium timidum, Parvimonas micra and Porphyromonas gingivalis [25,26]. Mechanistically, infections usually lead to gingival lesions with contamination of tissues surrounding the teeth [2,23]. Then, the lesion progresses to periodontitis once bacterial infection, and the subsequent inflammatory response, tackles the root surface, penetrating the supporting structures of the teeth [2,23] (Figure 1).

In general, the inflammation process begins with phagocytes (neutrophils and macrophages) that migrate to the lesion site. Importantly, this process is, at least in part, promoted by the gingival epithelium that releases chemical mediators including interleukins (ILs), prostaglandin E2 (PGE2), tumor necrosis factor alpha (TNF-α), that recruit neutrophils [27,28]. Furthermore, these phagocytic cells express on their plasma membrane specific receptors that recognize and bind surface molecules of bacteria (i.e., Toll-like receptors, TLRs) [27,28]. Analogously, the plasma proteins of the complement system react with one another to make pathogens more susceptible to the action of these phagocytic cells [28]. The function of this initial response includes the killing and elimination of microbes followed by an efficient clearance of the resulting cellular debris (necrotic tissue and apoptotic neutrophils) by mononuclear cells, such as monocytes and macrophages [28]. It is worth stressing that in an effective and healthy immune system, there is no damage to the tissue surrounding the tooth and the bacterial insult is efficiently removed [29,30]. However, when microbial species continue to grow, or if there is a defective/altered immune response, the acute periodontal inflammation becomes chronic and additional mediators are produced [28,30]. These events result in the recruitment of more immuno-cell types, such as T-cells and monocytes. Then, this prolonged inflammatory process induces alveolar bone reabsorption, by osteoclasts, and degradation of ligament fibers by MMPs, as well as the formation of the granulation tissue [31]. Moreover, as discussed above, this sustained chronic inflammatory process can lead to noxious effects that could link periodontal disease to other disorders including diabetes and CVD (Figure 1).

## 3. Diabetes and Periodontal Disease

Diabetes mellitus (DM) is a clinical syndrome, characterized by hyperglycemia, caused by inherited and/or acquired deficiency in insulin production and/or action [32,33]. Importantly, an association between DM and periodontitis was reported in the literature dating back to the 1960s. Since then, several reports have clearly demonstrated an association between DM and periodontal disease in both animals and humans [33]. For instance, in animal studies, Blasco-baque and coworkers recently provided data supporting the role of periodontal dysbiosis in the development of insulin resistance [34]. Moreover, Liu and coworkers demonstrated that periodontal disease aggravated pancreatic β-cell failure and insulin resistance in diabetic mice [35]. Importantly, evidence in humans has demonstrated that treating periodontal disease is able to reduce glycated hemoglobin in diabetic patients [36,37,38,39]. Furthermore, a recent observational study, performed in subjects aged 40 years or over, demonstrated that periodontitis was significantly more prevalent among individuals suffering from diabetes than non-diabetic ones, with no difference in terms of gender and age [40]. The specific mechanism connecting DM and periodontal disease has not been fully elucidated yet. Of note, several reports have suggested that DM participates in altering the subgingival bacterial community through substrate-related alterations offering a microenvironment favorable for the pathogens growth [41,42,43,44,45]. Furthermore, systemic levels of inflammatory mediators, including C-reactive protein (CRP), TNF-α, and IL-6, which are elevated in periodontal diseases, may represent the linchpin between DM and periodontitis [27,45,46,47,48,49]. For instance, in 2010 Chen Lei and coworkers provided data showing that periodontitis correlated with increased levels of glycated hemoglobin (HbA1c) and CRP observed in DM patients [49]. Moreover, in 2012 the same authors suggested that periodontal therapy was associated with CRP levels reduction in DM subjects [50]. In line with these reports, Quintero and colleagues have demonstrated that, periodontal therapy can also reduce HbA1c levels in subjects with DM [51]. Most importantly, data from a recent meta-analysis, by Grellmann and coworkers, supported the additional effect of the use of systemic antibiotics for subjects with diabetes and periodontitis [52]. However, in contrast with these data, a systematic review from Lira Junior and colleagues demonstrated that adjunctive use of systemic antibiotics did not provide any additional benefit in terms of the reduction of HbA1c in diabetic patients [53].

Finally, oxidative stress appears to be another major link between DM and periodontitis, since it can activate pro-inflammatory pathways common to these pathologies [54]. In this regard, Allen and colleagues have observed that DM, in patients with periodontitis, is associated with higher levels of plasma biomarkers of oxidative stress that may activate systemic pro-inflammatory pathways [55]. However, it is important to underline that several reports failed to show an association between diabetes and periodontal disease [56]. For this main reason, further investigations are warranted to confirm the potential link between these two highly prevalent disorders.

## 4. Periodontitis as a Risk Factor of Cardiovascular Disease

In the last two decades, several studies have showed that individuals with periodontitis are at higher risk of CV events, including, myocardial infarction, peripheral artery disease, stroke and heart failure (HF).

### 4.1. Myocardial Infarction

Myocardial infarction and periodontal disease share several common risk factors, including diabetes, smoking and inflammation [57,58,59]. For this reason, a growing body of evidence suggests that periodontal disease is associated with increased myocardial infarction risk [60,61,62,63]. Of note, in the 1980s Mattila and colleagues observed that dental health was significantly worse in patients with myocardial infarction than healthy controls [64]. Twenty-years later, Willershausen and coworkers demonstrated that there was a strict association between chronic dental infection and acute myocardial infarction [65]. Further to this, Jansson and coworkers suggested that oral disease could be used as a risk indicator of death due to CVD, especially when this was combined with other well-established risk factors, such as smoking [66].

Importantly, pre-clinical studies have also supported such a relationship. For instance, Akamatsu and colleagues demonstrated that, in mice, periodontal pathogens induced myocarditis and/or myocardial infarction [67].

However, since both myocardial infarction and periodontal disease are multifactorial in nature, some issues have been raised concerning the legitimacy of evidence sustaining such an association. Indeed, several epidemiologic studies failed to observe such a relationship [68,69]. Recently, one systematic review by Sidhu and coworkers concluded that no relationship between periodontal disease and myocardial infarction could have been exactly replicated or confirmed [70]. Therefore, since not enough evidence is available so far, the possibility that periodontal treatments are able to prevent the onset of myocardial infarction or the progression of post-myocardial infarction cardiac disease remains inadequately supported by factual evidence. Of note, this conclusion was congruent with a recent report from the American Heart Association [71]. Despite this, through the years, several studies continue to accumulate data regarding the presence of a specific relationship between periodontal disease and CVDs, including myocardial infarction, and potential mechanistic models have been proposed (discussed in the next paragraph) [72].

### 4.2. Endothelial Dysfunction

Endothelial dysfunction is an independent predictor of cardiovascular events and precedes the development of atherosclerosis and other CVDs [73]. This pathological process is usually caused by the reduced bioavailability of endogenous molecules, such as nitric oxide (NO^.^), a gasotransmitter that constrains platelet aggregation, inhibits the attachment of leukocytes to endothelial cells and prevents the expression of adhesion molecules [73,74]. Several lines of evidence have suggested a link between periodontitis and endothelial dysfunction. For instance, in 2008, Higashi and colleagues demonstrated that, in human subjects affected by periodontitis, endothelial dysfunction without cardiovascular risk factors or with hypertension, was due to a marked reduction in NO bioavailability and to systemic inflammation [75]. In line with this report, Moura and coworkers recently demonstrated a potential correlation between salivary NO concentration and endothelial dysfunction in patients with periodontal disease [76]. Furthermore, studies in experimental models of periodontitis in rats, have confirmed that reduced NO levels correlated with the onset of endothelial dysfunction [77,78]. Of note, a number of reports have also suggested that endotoxins and antigens secreted by periodontal bacteria can play an important role in the pathogenesis of endothelial dysfunction [79,80]. Moreover, it has been shown that periodontal bacteria are able to directly induce the up-regulation of several adhesion and chemoattractant molecules of endothelial origin that stimulates the attachment of leukocytes onto the surface of endothelial cells [81,82,83]. For instance, Porphyromonas gingivalis can induce a robust expression of the endothelial monocyte chemoattractant protein-1 (MCP-1) [84]. Similarly, Nakamura and colleagues demonstrated that the activation of Porphyromonas gingivalis, via lipopolysaccharide (LPS)-TLR2 system, mediated the adhesion of monocytes to endothelial cells [85]. In addition to these effects, Ansai and colleagues demonstrated that, in gingival epithelial cells, Porphyromonas gingivalis, can induce a significant rise in endothelin-1 expression and release [86]. Importantly, ET-1 acts as a vasoconstrictor factor and high levels of this molecule are associated with the onset of CVD [87]. In line with this body of evidence, it has been demonstrated that periodontal treatment alone or supplementation with antibiotics is able to improve endothelial dysfunction [88,89,90].

In this regard, a recent meta-analysis has demonstrated the beneficial effects of periodontal therapy on endothelial function [91]. More in detail, in their analysis, Orlandi and coworkers observed that periodontal disease was associated with greater carotid intima-media thickness (c-IMT) and with impaired flow-mediated dilation (FMD), indicating the presence of atherosclerosis and endothelial dysfunction, respectively. Nevertheless, in patients that have undergone intensive periodontal treatments, the analysis demonstrated a substantial improvement in the gingival condition associated with an increased FMD [91].

In line with these reports, Houcken and colleagues have demonstrated that periodontal disease is associated with increased arterial stiffness, as seen by augmented pulse wave velocity (PWV), therefore displaying higher atherosclerotic risk [92]. Importantly, while in this study, periodontal treatment did not result in reduced PWV [92] and in the report by Vidal and colleagues, periodontal therapy was able to attenuate arterial stiffness and reduce circulating inflammatory markers [93].

### 4.3. Peripheral Artery Disease (PAD)

PAD in its most advanced form, critical limb ischemia, represents a major health problem [94]. Importantly, a recent meta-analysis by Yang and colleagues demonstrated that PAD patients presented a higher risk of developing periodontitis compared to non-PAD subjects [95]. Moreover, these authors have observed that PAD patients have more missing teeth than non-PAD individuals. Importantly, this finding is in line with previous reports focused on this relationship. For instance, Chen and colleagues evaluated patients with PAD who underwent bypass surgery and reported that the majority of these patients presented periodontal bacterial infection (Porphyromonas gingivalis) of an anastomotic site of distal bypasses [96]. Interestingly, after adjusting for age, gender, smoking and DM, the authors found that periodontal disease increased up to five times the risk of developing PAD [96]. Furthermore, Ahn and coworkers [97] reported that patients with periodontitis had about a two-fold increase in the risk of PAD. In line with this observation, Çalapkorur and colleagues showed that periodontitis raised the odds ratio for developing PAD [98].

Importantly, due to the limited number of reports that have found a correlation between PAD and periodontitis, further studies with specific inclusion and exclusion criteria are required to confirm this relationship.

### 4.4. Stroke

Stroke is one of the most common causes of mortality worldwide [99]. Although, many risk factors have been identified as for the onset of this pathology, which includes pre-existing heart disease, hypertension, dyslipidemia, DM, smoking and age [99,100], several epidemiological studies (cross-sectional, cohort and case-control studies) have suggested periodontitis as a major potential cause of stroke [101]. Indeed, a number of recent meta-analyses and systematic reviews have demonstrated that the risk of cerebral ischemia and stroke is higher in subjects with periodontitis [17,18,102,103]. Of note, oral dysbiosis appears to be central in this association. For instance, Pussinen and colleagues observed that an elevated level of serum antibody to *Aggregatibacter actinomycetemcomitans* and *Porphyromonas gingivalis* correlated with stroke [104]. Moreover, Ghizoni and coworkers demonstrated that patients with stroke presented periodontitis at dental sites, with deep pockets, contaminated by *Porphyromonas gingivalis* [105]. Finally, similarly to other CVDs, Hosomi and colleagues found an association between serum C-reactive protein levels, and that the presence of antibody against Porphyromonas gingivalis is significantly associated with acute ischemic stroke [106].

### 4.5. Heart Failure

Heart failure (HF) is one of the leading causes of morbidity and mortality worldwide [99,107]. Importantly, its development is a consequence of many CVDs [99,107,108]. To date, only three studies have considered a potential relationship between periodontitis and HF [109,110,111]. In this regard, Frӧhlich and colleagues observed that HF-patients had a higher prevalence of periodontitis [109]. However, in their study, the authors found that the severity of periodontitis was not associated with HF etiology and symptom severity [109]. In contrast, Wood and Johnson suggested an association between these diseases [110]. In particular, they observed that patients with periodontal disease had a higher rate of HF development. Interestingly, these authors demonstrated that the monthly consumption of anti-oxidant and anti-inflammatory molecules contained in tomato, such as lycopene and carotenoids, exerted a protective effect in HF patients [110]. Finally, in a recent cross-sectional study, Schulze-Späte and colleagues found that HF was associated with periodontitis [111]. Indeed, HF patients exhibited more severe periodontitis that was associated with increased bone turnover markers compared with control patients. Interestingly, the authors suggested that local and systemic factors, including inflammatory mediators and citokines may account for this relationship [111].

## 5. Mechanistic Model for the Relationship Between Periodontal Disease and CVD

Although in clinical settings a potentially deep relationship between CVD and periodontitis has been observed, the detailed mechanism that connects these two pathologies has not been clarified yet. Nevertheless, oral pathogen dissemination into the bloodstream appears to be the major mechanism explaining such a relationship (Figure 1). For instance, bacteremia, often caused by non-surgical and surgical dental procedures, represents one of the major culprits of infective endocarditis in subjects predisposed to cardiac disease [112]. For this reason, prophylaxis is prescribed to patients with cardiac disease undergoing dental procedures [112].

Notably, periodontal pathogens can directly invade several organs and tissues, including the cardiovascular system. Accordingly, Louhelainen and colleagues recently reported that, in the pericardial fluids of patients with pericarditis, more than half (~60%) were positive for endodontitis-related bacteria while the remaining cohort (~40% of patients) were positive for periodontal pathogens [6]. Analogously, Oliveira and coworkers observed that periodontal pathogens were present in cardiac valve tissue of patients with valve disease [113]. Moreover, Ziebold and colleagues also demonstrated the presence of the DNA of oral bacteria in both atrial and ventricular tissues in patients that underwent aortic valve surgery [13].

Mechanistically, in pre-clinical studies, Sekinishi and coworkers demonstrated that, in mice undergoing transverse aortic constriction (TAC), the injection of Aggregatibacter actinomycetemcomitans (a periodontal pathogen) induced a significant cardiac function deterioration, compared to controls (TAC mice injected with PBS). This effect was accompanied by an augmented cardiac fibrosis and hypertrophy, and an enhanced atherosclerosis [114]. Interestingly, the Aggregatibacter actinomycetemcomitans infection resulted in a significantly increased MMP-2 expression in the interstitial tissue [114]. Importantly, MMPs are well recognized factors activated by periodontal pathogens, involved in both physiological tissue remodeling and in pathological extracellular matrix (ECM) degradation, mechanisms that are part of the pathogenesis of periodontitis [31].

Of note, other reports have demonstrated that periodontal pathogens are also able to invade arterial walls and colonize atherosclerotic plaques [115,116,117,118]. In particular, recent data demonstrated that Porphyromonas gingivalis, one of the major pathogens involved in periodontal disease, induced platelets aggregation and the expression of several cell adhesion molecules, such as the intercellular adhesion molecule 1 (ICAM-1), the vascular cell adhesion molecule 1 (VCAM-1) and p-selectin. Moreover, Li and colleagues reported that the protease Gingipain R, released from Porphyromonas gingivalis caused CVD by activating protein C, factor X and prothrombin. These mechanisms of action lead to aggregation of platelets and thrombotic clot formation [119]. Of note, systemic dissemination of periodontal pathogens and/or endotoxins can induce and inflammatory response, both locally and systemically. In this regard, once in circle, these microorganisms target large arteries, thus leading to the enhancement of vascular smooth muscle cells function, which represent one of the features of atherogenesis. Moreover, subsequent to bacterial infection, there is an augmented concentration of inflammatory mediators such as CRP [120], which has been proposed also as a potential risk factor for CVD development [121].

Finally, recent studies have indicated that chronic oral infection induces a high proportion of heat shock protein (HSP) 65 that increases the risk of CV events [122,123].

Despite all this evidence, more studies are needed, particularly in the experimental setting, to better understand and clarify the relationship between CVD and periodontitis.

## 6. Conclusions

In this review, we explored the potential association between periodontal disease and CVDs. Similarly, we described how periodontitis and diabetes were connected. Hence, the aim of this study was to increase the awareness among both clinicians and scientists about the need for a broader understanding of how periodontal disease prevention can impact CVD and diabetes. Doing so will not only provide additional preventative measures for CVDs, but also lower the economic burden on the health system. For this reason, effective health policy should focus on periodontitis as a CVD-related risk factor, due to the importance of preventing and treating all chronic infections.

Thus, in combination with diet, exercise and smoking control, preventing periodontal interventions should be in fact enlisted as an integral part of any adult health program directed to prevent, or more effectively manage CVD [124,125].

## Figures and Tables

**Figure 1 ijms-20-01414-f001:**
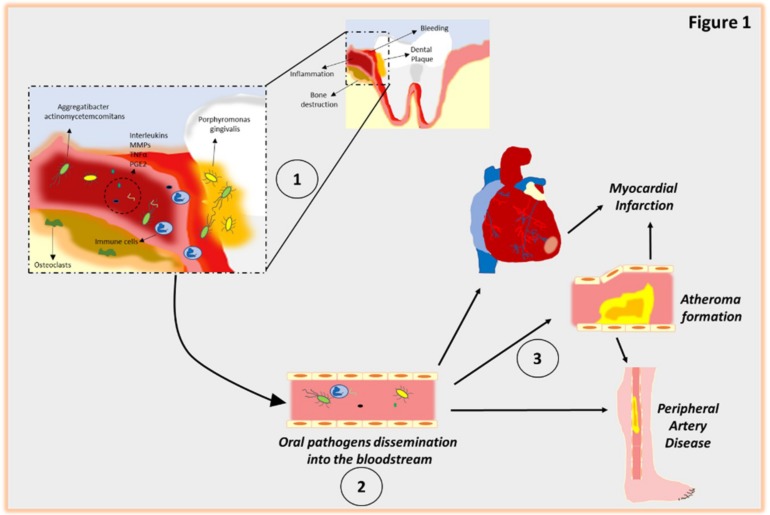
Schematic representation of inflammatory mechanisms linking periodontitis to CVDs. (1) The imbalance in pathogens of the biofilm leads to gingival epithelium inflammation that releases chemical mediators, such as interleukins (ILs), prostaglandin E2 (PGE2), tumor necrosis factor alpha (TNF-α) and MMPs, that recruit immune cells. This inflammatory response induces alveolar bone reabsorption, by osteoclasts. (2) At a chronic stage, oral pathogenic dissemination into the bloodstream leads to the onset of CVDs including atherosclerosis, myocardial infarction and peripheral artery disease (3).

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
