# Peer review of "Periodontal Disease: A Risk Factor for Diabetes and Cardiovascular Disease"

_ijms, 2019, doi:10.3390/ijms20061414_

Reviewer 1 Report

L20: change '..periodontitis acts on the same..' to '..periodontitis acts within the same..'.

L91: delete 'MMPs' as these do not per se 'recruit neutrophils'.

L130-134: The notion that 'Most importantly, data from a recent meta-analysis by Grellmann and coworkers supports an additional effect of the use of systemic antibiotics for subjects with diabetes and peritonitis' does no fit well with a 2017 meta-analysis indicating that adjunc systemic antibiotics provides no additional benefit on HbA1c in diabetics compared to nonsurgical periodontal treatment alone (Lira Junior R et al. J Dent 2017;66:1-7).

L220: change 'patients with PAD underwent..' to '..patients with PAD who underwent..'

L223: change 'Authors' to 'authors'.  

Author Response

Response to Reviewer 1 Comments

We would like to thank the Reviewer for the favorable remarks and constructive criticism that helped us to improve our manuscript.

Point 1: L20: change '..periodontitis acts on the same..' to '..periodontitis acts within the same..'.

Response 1: We have modified the text as indicated. Thanks.

Point 2: L91: delete 'MMPs' as these do not per se 'recruit neutrophils'.

Response 2: Deleted as suggested. Thanks.

Point 3: L130-134: The notion that 'Most importantly, data from a recent meta-analysis by Grellmann and coworkers supports an additional effect of the use of systemic antibiotics for subjects with diabetes and peritonitis' does no fit well with a 2017 meta-analysis indicating that adjunc systemic antibiotics provides no additional benefit on HbA1c in diabetics compared to nonsurgical periodontal treatment alone (Lira Junior R et al. J Dent 2017;66:1-7).

Response 3: We apologize for this missing information. As suggested by the Reviewer we referenced the study from Lira Junior et al.. Please see page 4 line 52. Thanks.

Point 4: L220: change 'patients with PAD underwent..' to '..patients with PAD who underwent..'

Response 4: We have modified the text as indicated. Thanks.

Point 5: L223: change 'Authors' to 'authors'. 

 Response 5: We have modified the text as indicated. Thanks. 

Reviewer 2 Report

In the manuscript entitled, “Periodontal disease: a risk factor for diabetes and cardiovascular disease,” by Liccardo et al, the authors provide a thorough review of the connection between periodontal disease, diabetes and cardiovascular disease. The authors to a good job of covering the literature pertaining to the subject and there are only a few minor, typographical corrections that need to be made.

Page 1 line 34, the “highly” seems out of place

Page 2 line 48, the “raise” should be rise

Page 2 line 49, “in disparate tissues and organs of the cardiovascular system,” would be more appropriate

Page 2 line 51, “For these reasons, in the last decades,” would be more appropriate

Page 3 line 81, “periodontitis” is capitalized when it shouldn’t be.

Page 3 line 91, there should be an “and” between TNF-α and MMPs

Page 3 line 97, neutrophils is spelled incorrectly

Page 4 line 149, an abbreviation for myocardial infarction is given but it is never used in the text. It should either be removed or myocardial infarction should be referred to as MI for the remainder of the document.

Page 5 line 167, “no” should be changed to “not” so it reads, “the progression of post-MI cardiac disease remains not adequately supported…”

Page 5 line 176, I feel a word is missing in, “a reduced bioavailability of endogenous, such as…” Maybe make it “endogenous compounds” or “endogenous molecules”.

Page 5 line 190, commas aren’t needed around, “of endothelial origin”

Page 5 line 198, it should be, “In line with this evidence” or, “in line with these evidences”

Page 5 line 199, periodontal shouldn’t be capitalized

Page 5 line 205, it should be, “Nevertheless, in patients that have undergone…”

Page 6 line 216, no comma is needed after meta-analysis

Page 6 line 220, some words have been omitted. It should read, “have evaluated patients with PAD that underwent bypass surgery and reported”

Page 6 line 223, page 7 line 250, page 7 line 259, authors is capitalized when it shouldn’t be

Page 7 line 233, there is a parenthesis after age that doesn’t need to be there

Page 7 line 265, pathogens should be pathogen

Page 7 line 266, culprit should be culprits

Page 7 line 272, the was after “(~40%) of patients)” should either be omitted or changed to were

Author Response

Response to Reviewer 2 Comments

We are thankful to the Reviewer for his/her appreciation of our work and for the constructive feedback.

Point 1: Page 1 line 34, the “highly” seems out of place

Response 1: Deleted as suggested. Thanks.

 Point 2: Page 2 line 48, the “raise” should be rise

 Response 2: We have modified the text as indicated. Thanks.

 Point 3: Page 2 line 49, “in disparate tissues and organs of the cardiovascular system,” would be more appropriate

 Response 3: We have modified the text as indicated. Thanks.

 Point 4: Page 2 line 51, “For these reasons, in the last decades,” would be more appropriate

 Response 4: We have modified the text as indicated. Thanks.

  Point 5: Page 3 line 81, “periodontitis” is capitalized when it shouldn’t be.

 Response 5: We have modified the text as indicated. Thanks

Point 6: Page 3 line 91, there should be an “and” between TNF-α and MMPs

Response 6: We have modified the text as indicated. Thanks.

 Point 7: Page 3 line 97, neutrophils is spelled incorrectly

Response 7: We have modified the text as indicated. Thanks.

 Point 8: Page 4 line 149, an abbreviation for myocardial infarction is given but it is never used in the text. It should either be removed or myocardial infarction should be referred to as MI for the remainder of the document.

 Response 8: MI was deleted as indicated. Thanks.

 Point 9: Page 5 line 167, “no” should be changed to “not” so it reads, “the progression of post-MI cardiac disease remains not adequately supported…”

 Response 9: We have modified the text as indicated. Thanks.

 Point 10: Page 5 line 176, I feel a word is missing in, “a reduced bioavailability of endogenous, such as…” Maybe make it “endogenous compounds” or “endogenous molecules”.

 Response 10: We have added “molecules” as indicated. Thanks.

 Point 11: Page 5 line 190, commas aren’t needed around, “of endothelial origin”

 Response 11: We have modified the text as indicated. Thanks.

 Point 12: Page 5 line 198, it should be, “In line with this evidence” or, “in line with these evidences”

 Response 12: We have modified the text as indicated. Thanks.

 Point 13: Page 5 line 199, periodontal shouldn’t be capitalized

 Response 13: We have modified the text as indicated. Thanks.

 Point 14: Page 5 line 205, it should be, “Nevertheless, in patients that have undergone…”

 Response 14: We have modified the text as indicated. Thanks.

 Point 15: Page 6 line 216, no comma is needed after meta-analysis

 Response 15: We have modified the text as indicated. Thanks.

 Point 16: Page 6 line 220, some words have been omitted. It should read, “have evaluated patients with PAD that  underwent bypass surgery and reported”

 Response 16: accordingly to the suggestions of this Reviewer and Reviewer 1 we have modified the text as follow: evaluated patients with PAD who underwent bypass surgery and reported

 Point 17: Page 6 line 223, page 7 line 250, page 7 line 259, authors is capitalized when it shouldn’t be

 Response 17: We have modified the text as indicated. Thanks.

 Point 18: Page 7 line 233, there is a parenthesis after age that doesn’t need to be there

 Response 18: Deleted as suggested. Thanks.

 Point 19: Page 7 line 265, pathogens should be pathogen

 Response 19: We have modified the text as indicated. Thanks.

 Point 20: Page 7 line 266, culprit should be culprits

 Response 20: We have modified the text as indicated. Thanks.

 Point 21: Page 7 line 272, the was after “(~40%) of patients)” should either be omitted or changed to were

 Response 21: We have changed “was” to “were” as indicated. Thanks.

This manuscript is a resubmission of an earlier submission. The following is a list of the peer review reports and author responses from that submission.

Round  1

Reviewer 1 Report

This topic is interesting topic for periodontist.

In introduction, gums, collagen fibers should be replaced by periodontal ligament, cementum.

Aggregatibacter is correct.

Microbes should be expressed by italic style.

 This manuscript should be give a full explanation.

Reviewer 2 Report

This is an important topic that is worthy of a review. It remains to be seen in diabetes which is more relevant, diabetes causing periodontal infections or infections worsening diabetes (most likely). The authors should look to include literature evaluating the effect of antibiotic use to reduce CVD or diabetes severity in people with periodontal disease.

Reviewer 3 Report

 Periodontal disease: a risk factor for diabetes and cardiovascular disease

Comments to the authors

Major concerns

Periodontal disease (PD) and atherosclerosis are chronic inflammatory diseases, and the bulk of evidence for an association between with PD and CVD is for atherosclerotic disease, namely myocardial infarction (MI, which is by far the most well-established CVD linked to PD), stroke and peripheral artery disease (PAD). Therefore, it is surprising that the Abstract highlights ‘myocardial hypertrophy’ (L22) and that MI only comes in as no. 3 CVD (L172, after PAD and heart failure) in the central text. Inflammation may contribute to virtually all diseases including most types of CVD, but the link with PD (from bench to bedside) is simply much better for atherosclerotic disease.  Therefore, to me, the author’s entire omission of literature on PD-stroke, and, on the other hand, inclusion of letter weight signals in the literature about PD and myocardial hypertrophy, HF (without mentioning that HF is mostly caused by coronary atherosclerotic disease), myocarditis (L71), pericarditis (L203), and bacteria in cardiac valves (endocarditis?, L205), respectively, is disturbing.  Along this line, it is disappointing that extensive literature on an association between PD (and its treatment) and endothelial function, i.e. the precursor of atherosclerotic disease (and, for that matter the link between PD and carotid intima-media thickness or arterial stiffness, both also subclinical markers of atherosclerosis) is completely absent. Also, the as yet only available data linking PD treatment to less CVD should be cited (Chen Z-Y et al.  Am J Med 2012;125:568-75; Chou S-H et al. PLoS ONE 2015 Jun 26;10(6):e0130807), along with the few (but potentially seminal) reports indicating that bacterial DNA detected in atherosclerotic plaques actually represent viable bacteria (not just DNA in dead bacteria engulfed by leukocytes present in plaques, which by the way is a possibility that the authors ignore) and be organized in biofilms, the latter possibly being the reason that trials of antibiotic therapy have failed in CVD (Kazarov EV et al. ATVB 2005;25:e17-18, Lanter BB et al. mBio 2014;5:e01206-14 and Infect Immun 2015;83:3960-71, and Snow DE et al. J Wound Care 2016;25:516-22).

Furthermore, the stated aim to provide updated evidence on the link between PD and CVD (L25) seems at odds with a parallel discussion of the PD-diabetes link that does not really focus on CVD (L106-136). Epidemiological data in the literature linking PD to MI and stroke were, of course, adjusted for diabetes, so irrespective of the interesting bi-directional relation between PD and diabetes (that actually also exists between PD and rheumatoid arthritis or PD and psoriasis, respectively, and is likely related to inflammatory pathways that are also shared by atherosclerosis) incremental CVD risk associated with diabetes in patients with PD may actually be less clear. Also, it seems symptomatic to the quality of the literature search that there are at least 3 more recent meta-analyses of effects of periodontal treatment on glycemic control than the one from 2013 used by the authors ref. 35, e.g. Teshome A,  Yitayeh A. BMC Oral Health 2016;17(1):31, Faggion CM Jr et al. J Periodontal Res 2016M51:716-25, Artese HP et al. PLoS ONE 2015;10(5):e0128344).

The text has plenty of linguistic problems, including fluffiness of terms like ‘plausible to hypothesize’ (L19), ‘reasonable to hypothesize’ (L23), ‘more than possible to speculate’ (L48), ‘sensible increased risk’ (L54), ‘sensitize clinicians’ (L244), ‘health payback model’ (L245) etc.]. Also, some references are misplaced, e.g. ref. 55 and 56 cannot reasonably support both that periodontitis raises the odds for developing PAD (L52) and that HF is a consequence of many CVDs’ (L59) AND at the same time, that periodontal pathogens are able to invade arterial wall and colonize atherosclerotic plaques (L219).